# Graph-based Classification of Intestinal Glands in Colorectal Cancer Tissue Images

Linda Studer[*,1,2], Shushan Toneyan[*,3], Inti Zlobec[3], Heather Dawson[3], and Andreas Fischer[1,2]

[1] DIVA Research Group, University of Fribourg
`firstname.lastname@unifr.ch`
[2] Institute of Complex Systems (iCoSys)
University of Applied Sciences and Arts Western Switzerland
`firstname.lastname@hefr.ch`
[3] Institue of Pathology, University of Bern,
`firstname.lastname@pathology.unibe.ch`

**Abstract.** Pathologists study tissue morphology in order to correctly diagnose diseases such as colorectal cancer. This task can be very time consuming, and automated systems can greatly improve the precision and speed with which a diagnosis is established. Explainable algorithms and results are key to successful implementation of these methods into routine diagnostics in the medical field. In this paper, we propose a graph-based approach for intestinal gland classification. It leverages the high representational power of graphs for describing geometrical and topological properties of the glands. A novel, publicly available image and graph dataset is introduced based on cell segmentation of healthy and dysplastic H&E stained intestinal glands from pT1 colorectal cancer. The graphs are compared using an approximate graph edit distance and are classified using the k-nearest neighbours algorithm. With this method, we achieve a classification accuracy of 83.3%.

**Keywords:** Intestinal Gland Classification · Colorectal Cancer · Digital Pathology · Graph Matching · Graph Edit Distance.

## 1 Introduction

In diagnosis and treatment of colorectal cancer the observations of pathologists are crucial for the characterisation of the stage of the disease and subsequent predictions of its progression [18]. The precise morphological characteristics of the deformation depend on the type of cancer and the stage of cancer progression [4]. In order to expedite diagnostics and reduce errors and variability between experts, computer-aided diagnosis (CAD) can offer great support in the diagnostic process.

In the initial stages of cancer, such as pT1, carcinomas can frequently be observed originating from polyps [4]. In such cases it is possible to observe normal

---

[*] These authors contributed equally to this work.

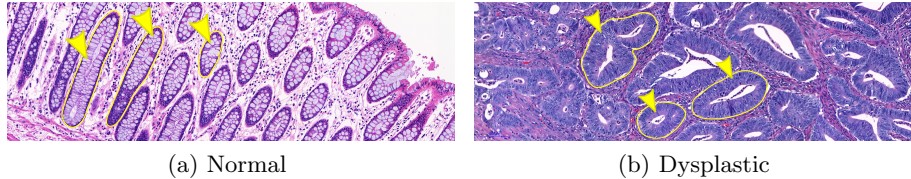

(a) Normal     (b) Dysplastic

*Fig. 1: Examples of normal (a) and dysplastic (b) colon mucosa stained with H&E. The yellow arrowheads point to selected glands. Normal glands usually have a regular round or oval shape while dysplastic glands are more irregularly shaped.*

tissues, dysplasia and carcinoma on one slide. In normal tissue the glands have parallel and flat lumina lined with a single layer of cells in the upper parts of the gland. They have circular or oval shape and a homogeneously coloured nucleus, that is pushed to the outer side of the gland by the mucus in the cytoplasm. In dysplastic glands the regular and ordered configuration is disrupted, and their shape varies greatly, especially when a mix of different dysplasia types (such as low- and high-grade) is present.

Histological images show the microanatomy of a tissue sample. They contain many different cell types, cell compartments and tissues which makes them very complex to analyse. In their diagnosis, pathologists consider morphological changes in tissue, spatial relationship between cell (sub-)types, density of certain cells, and more. Graph-based methods, which are able to capture geometrical and topological properties of the glands, offer a very natural approach to attempt an automated analysis of such data [15].

Graphs have been used for a variety of tasks in digital pathology, such as classification and exploratory analysis [15] as well as segmentation [17] and content-based image retrieval (CBIR) [14]. There are also a great variety of types of graphs that are being used, such as O'Callaghan neighbourhood graphs, attributed relational graphs (ARG) and cell graphs [15]. Especially cell graphs have been successfully used to support cancer diagnosis [3,9].

In this paper, we propose a gland classification method based on labelled cell graphs and graph edit distance (GED) [6,7], which transforms one cell graph into another using deletion, insertion, and substitution of individual cells. In contrast to other graph matching methods [5], such as spectral methods [11] or graph kernels [8], GED has the advantage that it is applicable to any type of labelled graphs. Furthermore, it provides an explicit mapping of cells from one gland to cells of the other gland, which may help human experts comprehend why the algorithm predicts high or low gland similarity (see for example Figure 3). For experimental evaluation, we have created a graph dataset that contains cell graphs of healthy and dysplastic H&E stained intestinal glands from pT1 colorectal cancer. The dataset, which has been made publicly available, is used to evaluate the classification performance of our GED-based approach using a k-nearest neighbour (k-NN) classifier. The performance is compared to results reported in the literature.

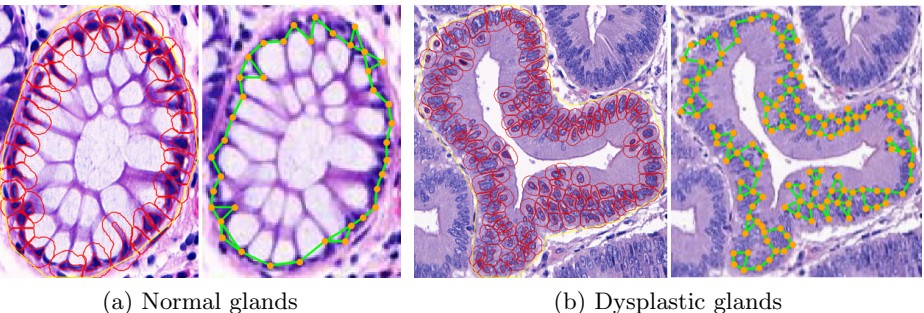

(a) Normal glands                    (b) Dysplastic glands

*Fig. 2: Examples of the cell segmentation (left) and graph-representation (right) of a normal and a dysplastic gland. Each detected cell (circled in red) is represented as a node in the graph (in orange). The nodes are connected with edges (in green) based on the physical distance between them.*

## 2   Graph-Based Gland Classification

This section introduces the novel, publicly available gland classification dataset and provides more details about the proposed method for graph-based gland classification.

### 2.1   pT1 Gland Graph (GG-pT1) Dataset

The images used to create the graphs dataset are from H&E stained whole slide images (WSIs) tissue samples taken from pT1 [4] cancer patients. The glands are cropped from images that have normal tissues, dysplasia and carcinoma on one slide. The crops are classified by an expert pathologist and then used to build the graphs. In total there are 520 graphs from one 20 different patients. One WSI per patient was selected based on image quality and 26 well-defined glands (13 dysplastic and 13 normal) were manually annotated.

The cells of each gland are segmented using QuPath [1]. The same parameters are used for all the images. 33 features are exported from QuPath for each cell, which are used to label the nodes. Available features based on the cell are the eosin stain (mean, standard deviation (SD), min, max), circularity, eccentricity, perimeter, area and diameter (min, max). Features based on the nucleus are circularity, eosin stain (mean, std, min, max, range, sum), hematoxylin stain (mean, std, min, max, range, sum), diameter (min, max), area, perimeter and eccentricity. Further features are the eosin stain (mean, std, min, max) of the cytoplasm and the nucleus/cell area ratio.

Figure 2 shows example images and graphs from the dataset which is publicly available[1]. It includes all images, annotation masks, and graph features as well as the reference, validation and test split used in this paper.

---

[1] https://github.com/LindaSt/pT1-Gland-Graph-Dataset

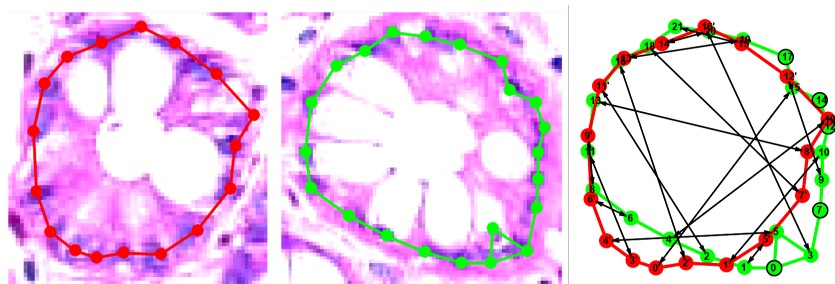

Fig. 3: GED transformations between two normal glands. The black arrows indicate node label substitution, the nodes circled in black mark deleted/inserted nodes.

## 2.2 Graph-Based Representation

The formal mathematical definition of a graph $G$ is given as a tuple of $(V, E, \alpha, \beta)$, where $V$ is the finite set of nodes (or vertices), $E$ is the set of edges, $\alpha$ is the node labelling function and $\beta$ is the edge labelling function.

We use so-called cell graphs [9], in which each cell is represented by a node with different attributed features $\alpha(v) = (x_1, \ldots, x_n)$ with $n \leq 33$ (see section 2.1 for the complete feature list). Figure 2 shows examples of cell graphs of glands. Because the different features all have a different range, they are normalised using the z-normalisation which adjusts each feature value $x$ such that $\hat{x} = \frac{x-\mu}{\sigma}$.

For each node, we insert an edge to its two spatially closest neighbour nodes. No edge features are used. Node features are selected using the sequential forward selection method [10]. This process starts with no features and iteratively adds the best feature until there is no further improvement in the classification accuracy. We also establish a baseline based on the unlabelled graph.

## 2.3 Graph Edit Distance (GED)

GED is an error-tolerant measurement of similarity between two graphs [7]. It provides a model for transforming a source graph into a target graph instead of searching for an exact match between graphs or their sub-graphs. Figure 3 shows an example of such a transformation.

GED is defined as the distance between two graphs in the case when the cost of transforming one into the other is minimal. There are three types of edit operations that are performed on edges as well as labels to transform a graph: insertion, deletion and label substitution. For each of these operations a cost function needs to be specified. We consider the Euclidean cost model that uses a fixed cost for deletion/insertion and the Euclidean distance for substituting node labels. Since we do not use edge labels in our cell graphs, the cost function for edge label substitution does not need to be defined.

The computational complexity of the exact GED calculation increases exponentially as a function of the number of nodes in the graphs. However, heuristic methods are available that can compute an approximate solution. We use an

*Table 1: Classification accuracies achieved by the baseline and after forward search selection of the node features. The mean accuracy along with the standard deviation of a 4-fold cross-validation is reported.*

|  | NODE FEATURES | ACCURACY |
|---|---|---|
| BASELINE | NONE | $71.7 \pm 2.8\%$ |
| OPTIMIZED GRAPH | CYTOPLASM: EOSIN MIN
NUCLEUS: HEMATOXYLIN MEAN, MIN, MAX | $83.3 \pm 1.7\%$ |

improved version of the bipartite graph-matching method (BP2) [6], which runs in quadratic time and calculates an upper bound of GED.

### 2.4 K-Nearest Neighbour Classification

The classification of the glands is performed using the k-NN classifier, which assigns the most frequent label out of the $k$ most similar objects to the object to be classified [2]. In our case we use the three closest ($k = 3$) gland graphs in the reference set, in terms of the GED, to classify a new gland graph.

## 3 Experimental Evaluation

Our goal is to classify intestinal glands in the novel GG-pT1 dataset as either normal or dysplastic by using graph-based representations. Graphs are compared to a reference dataset using the GED and then classified using the k-NN algorithm. We also investigate the impact of node feature selection.

### 3.1 Setup

We split the dataset into four parts and evaluate the performance with a 4-fold cross-validation. Two parts are used as the reference set and one each for the validation and test set (details here[2]). The reference set is used for the classification. For each input graph the GED is computed to all graphs in the reference set and k-NN is then used to classify the graph based on this distance. The validation dataset is used to optimise the insertion/deletion cost for nodes and edges using a grid search over 25 parameters (for the specific parameters see here[2]) and to optimise the node features using forward search.

### 3.2 Results

Table 1 gives an overview of the results and selected node features. The achieved accuracy is 83.3%. The forward search selected four attributes, three are based on

---

[2] https://bit.ly/2xDuRcV

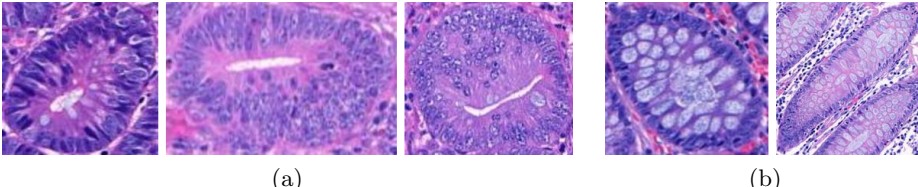

(a)                                                                                          (b)

*Fig. 4: Examples of dysplastic glands misclassified as normal (a) and normal glands misclassified as dysplastic (b).*

the nucleus hematoxylin stain and one is based on the cystoplasm eosin stain. Adding these four features to the nodes increased the performance by 11.6% compared to the unlabelled baseline.

### 3.3   Discussion

We achieve slightly better results on our dataset than the only other published results using graph-based methods on a closely related (but not non-publicly available) colorectal cancer image dataset [13]. Ozdemir et al. report results achieved by hybrid models that use different combinations of structural and statistical features. One variant uses GED embedding coupled with a Support Vector Machine (SVM) to classify glands as normal, low- or high-grade dysplastic and achieves an overall accuracy of 81.72%. They however use a different graph-representation. Their graphs are not based on cells, but they identify nodes as circular nucleus and non-nucleus objects and label them with some additional features based on the expansion order in a breadth-first search.

The precision is higher among the normal glands (see Figure 4 for examples). Looking at the misclassified dysplastic glands, most of them show features that are very distinct for dysplastic glands such as structural chromatin and nuclear polarity, which are currently not available as node labels. Adding these features could thus improve the classification accuracy. Many of them also have a more round shape, which is more similar to the shape of normal glands. Some of the graphs also show some issues with cell segmentation, which reduces the representational power of the graph. A few of them are very low-grade dysplasia and thus have very similar features to normal glands. Improving the cell segmentation method and adding more key features could overcome these misclassification errors. Extending the dataset should also improve the performance. For an accurate classification, it is very important that the reference set contains a strong representation of the different slicing planes and varieties in appearance.

Figure 3 shows an example of the matching of two graphs. We can see which individual cells from each graph are matched by label substitution and therefore have similar local features. Some cells are not matched and are inserted/deleted during the transformation because they are too different from any of the cells in the other graph. This illustration of the matching has the potential to help

humans better understand the result of the automatic classification and thus to help with explainability.

To create this dataset, the gland selection was performed manually. For this system to be useful in routine diagnostics, the gland detection and segmentation process should be automatised. This is another focus for future work.

## 4    Conclusion

In this preliminary study we achieve an accuracy of 83.3% for intestinal gland classification (normal or dysplastic) using a graph-representation with distance-based edges and four node features coupled with a GED and k-NN classification. This result is comparable to state-of-the-art results for graph-based gland classification reported on a private dataset [13].

There are a number of possibilities to further improve the classification results presented here. On the graph extraction level, improving the cell segmentation helps to create more precise graph-representations. There is also a vast range of different graph types that can be explored that include more tissue types and areas, such as the lumen of the glands, many of which have already been successfully used to analyse histopathological data [15]. Exploring different features for the nodes and edges can also help to improve the performance. There is a wide range of possibilities here and the GED is well suited for this task, as it is able to handle any type of labelled graph. Using a different classifiers such as SVMs [13] or even combining different types of graph-representations and classifiers into an ensemble may also lead to a higher accuracy. Another option and one of the newest techniques for graph classification is geometric deep learning, which is based on graph neural networks [12].

Future work also includes more experimental evaluations, such as on the publicly available Gland Segmentation Challenge Contest (GlaS) [16] dataset, which is an image dataset of intestinal glands from colorectal cancer tissue. So far we have not been able to obtain a useful cell segmentation on this dataset with our methods, as the images are of much lower resolution than in our dataset.

Furthermore, we plan to conduct a study with expert pathologists to evaluate what kinds of graphs and graph matching methods are most intuitive and understandable in order to improve the explainability of our method. We also want to establish an expert pathologist baseline to analyse the inter-observer variability and include the experts knowledge into the feature selection process.

## Acknowledgment

The work presented in this paper has been partially supported by the Rising Tide foundation with the grant number CCR-18-130.

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
