# OpenReview forum: "Graph-based Classification of Intestinal Glands in Colorectal Cancer Tissue Images"
_MICCAI.org/2019/Workshop/COMPAY — COMPAY 2019_

### Official Review · AnonReviewer1 · 2019-07-28
**I recommend acceptance, after a few editings/clarifications.**

**Rating:** 7
**Confidence:** 4

**Review:**

The authors propose a graph-based method to classify intestinal glands into benign or diseased. The paper is well written and understandable. The pipeline uses adjusted versions of some existing methods. One major contribution of the authors is the provisioning of a new dataset with segmented colon gland images. Their classification accuracy is 83.1% in their dataset.

However, here are some concerns which I raise:

- The methodology seems to be a (maybe new) application of existing methods in graph-based CV. GED, KNN, standard HE image features, are all existing. The authors should state more clearly if and what their new methodology contribution is.

- At the same time, they provide a new dataset consisting of 520 gland images plus graphs, from 20 different WSI (13 normal/13 diseased glands per WSI). Although this is a step in the right direction, I think this dataset should be improved to be more relevant. Why only 20 WSI? And which 13 glands per class and image? Who selected them according to which criterium? Do you think 520 glands reflect the high variability of deformed glands in cancer? I highly doubt it.

- Please describe the dataset better: What is the magnification, mu/px? What is the dimension range of the image crops? What is the range of number of nodes in the graphs? E.g. here you can split these statistics by healthy and diseased.

- Is the number of nodes (absolute or normalized by circumference) alone not already a good classifier of diseased glands? Fig 2 implies so. You could include this or are simple Random Forest with the statistics-features in your baseline in Table 1.

- Is the classification of glands for pathologists easy? What is the inter-pathologist error?

- With such a small dataset, I would expect image normalization (intensity, color). Any applied?

- How have the cells been detected? Manually? Via Qu-Path? If via Qu-Path, was that one pipeline for all images, or adjusted per image to "give best results"?

- QuPath has been used to segment the images. How? One pipeline for all images? How have the parameters been set? I would be nice to see some example segmentations, since these are the basis for the feature extraction.

- For each node, the authors inserted and edge to its two spatially closest neighbors. Does it never happen that the graph circle is not closed? Doesn't that even matter?

- My biggest concern in the manuscript is the validation part: The dataset has been split into reference, validation and test. How? On WSI basis? On patient basis? Or randomly on gland basis (this would introduce correlation between reference and test)?

- Additionally, to show robustness of your model, I would expect to repeat the split several times differently (cross-validation?). Is 83.1% robust? Significant? Or just a lucky split of training and test?

- The authors discuss their error modes (FN or FP...). However, they should consider that FP are "better" as the pathologist can correct (while FN are potentially missed cancers). Unfortunately, their method produces more FN than FP. The authors could discuss what could be done to adjust the model such that the FN rate goes down (increasing the FP rate).

- The paper somehow refers to the Figures in more or less random order. If possible, please restructure.

- Fig 1b and Fig 4 are hard to see when printed.

---

### Official Review · AnonReviewer4 · 2019-08-15

**Rating:** 6
**Confidence:** 3

**Review:**

This paper presents a method to classify colon glands based on a graph representation of segmented cells. Cell segmentation is performed using a publicly available software package (QuPath), and glands are manually pre-selected. A kNN classified is used to compare graphs at test time with a set of reference graphs, achieving performance of >80% accuracy. Authors have made the used dataset publicly available, which is a positive effort towards reproducible science.

Comments:

•	The authors should show, or at least discuss, how achieved performance depend on the used classifier (i.e., kNN). A comparison with other classifiers, such as SVM or Random Forests would have been a useful to get insights on the role of the classifier in this framework.
•	The paper lacks a proper comparison with state of the art, as the comparison reported is solely based on values reported in another paper, which uses another dataset, therefore not directly comparable.
•	Glands were cropped manually and checked by pathologists. then classified using this algorithm. What is the clinical applicability of this system? Is the interaction of a human observer always needed to select glands? In this context, the authors should discuss in their paper what is the advantage of using classification instead of (instance) segmentation and classification, as done by other researchers.
•	Have the authors considered to use extracted features to describe the patch itself, without using graphs, and have a model classify those features directly? What would be the disadvantage of this approach, or in other words, what is the main advantage of using a graph-based representation?

---

### Decision · Program_Chairs · 2019-08-20

Accept